# A Multiomic Analysis of Chicken Serum Revealed the Modulation of Host Factors Due to *Campylobacter jejuni* Colonization and In-Water Supplementation of Eugenol Nanoemulsion

**DOI:** 10.3390/ani13040559

**Published:** 2023-02-05

**Authors:** Basanta R. Wagle, Austin Quach, Seungjun Yeo, Anna L. F. V. Assumpcao, Komala Arsi, Annie M. Donoghue, Palmy R. R. Jesudhasan

**Affiliations:** 1ASRT Inc., Centers for Disease Control and Prevention, Atlanta, GA 30333, USA; 2Dalton Bioanalytics, California Nanosystems Institute, Los Angeles, CA 90095, USA; 3Department of Poultry Science, University of Arkansas, Fayetteville, AR 72701, USA; 4Poultry Production and Product Safety Research Unit, Agriculture Research Service, United States Department of Agriculture ARS, USDA, Fayetteville, AR 72701, USA

**Keywords:** *Campylobacter*, multiomic analysis, eugenol, phytochemicals, broiler chicken, foodborne pathogen, LCMS-based technology

## Abstract

**Simple Summary:**

*Campylobacter jejuni,* a normal flora in the chicken gut, is one of the most common causes of bacterial gastroenteritis in humans, affecting approximately 95 million people worldwide. Carcass contamination at the processing plant through the leakage of gut contents can lead to foodborne illnesses when contaminated poultry or poultry products are consumed or improperly handled. This study focused on identifying host factors modulated by *Campylobacter* colonization in untreated chickens and those treated with phytochemical eugenol. We used a novel LCMS-based multiomics technology to identify the factors modulating the colonization of *C. jejuni* in broiler chickens. Three groups of broiler chickens were used: (1) negative control, (2) positive control, and (3) eugenol nanoemulsion (EGNE) treatment–supplemented with 0.125% EGNE in the water. Based on the multiomic analysis (proteins, lipids, and metabolites), we identified a few key host factors that were modulated with the colonization of *C. jejuni*.

**Abstract:**

*Campylobacter jejuni* is a foodborne pathogen that causes campylobacteriosis globally, affecting ~95 million people worldwide. Most *C. jejuni* infections involve consuming and/or handling improperly cooked poultry meat. To better understand chicken host factors modulated by *Campylobacter* colonization, we explored a novel LCMS-based multiomic technology using three experimental groups: (1) negative control, (2) positive control, and (3) eugenol nanoemulsion (EGNE) treatment (supplemented with 0.125% EGNE in the water) of broiler chickens (n = 10 birds/group). Birds in groups two and three were challenged with *C. jejuni* on day 7, and serum samples were collected from all groups on day 14. Using this multiomic analysis, we identified 1216 analytes (275 compounds, seven inorganics, 407 lipids, and 527 proteins). The colonization of *C. jejuni* significantly upregulated CREG1, creatinine, and 3-[2-(3-Hydroxyphenyl) ethyl]-5-methoxyphenol and downregulated sphingosine, SP d18:1, high mobility group protein B3, phosphatidylcholines (PC) P-20:0_16:0, PC 11:0_26:1, and PC 13:0_26:2. We found that 5-hydroxyindole-3-acetic acid significantly increased with the EGNE treatment when compared to the positive and negative controls. Additionally, the treatment increased several metabolites when compared to the negative controls. In conclusion, this study revealed several potential targets to control *Campylobacter* in broiler chickens.

## 1. Introduction

Human campylobacteriosis is the major foodborne illness caused by *Campylobacter* and is responsible for more than one million cases and economic losses of over $1.9 billion annually in the United States [1]. About 90% of human campylobacteriosis is caused by *Campylobacter jejuni*, a species of *Campylobacter* commonly present in chickens’ intestinal tract [2,3,4]. *C. jejuni* colonizes the chicken intestinal tract in early life and persists through slaughter. Worldwide, 50% of poultry meat is contaminated with *Campylobacter* during slaughter and carcass processing, effecting workers and consumers who handle raw poultry meat and posing a unique hazard to those who consume undercooked chicken meat [4].

Controlling *Campylobacter* contamination in poultry and poultry products at production level would enhance the safety of poultry products for human consumption. The effective control of *Campylobacter* in preharvest poultry would significantly reduce the incidence of human foodborne illnesses as well as economic losses due to campylobacteriosis [5].

The genetic factors of *C. jejuni* that mediate attachment and colonization in the ceca of chickens are well studied [6]. However, the response of chicken host immune factors, including proteins, lipids, and other metabolites in serum that, modulated with *C. jejuni* colonization, is relatively unexplored. In addition, our laboratory has been investigating the potential use of essential oils for controlling *C. jejuni* for a decade and reported their antibacterial effects on *C. jejuni* colonization in broiler chickens [7,8,9,10,11,12,13]. Previously, we reported that in-water supplementation of phytochemicals significantly reduced the *C. jejuni* counts in broiler chickens and investigated the underlying mechanism of antibacterial actions. However, the modulation of host immune factors upon colonization and with treatment has yet to be determined. 

To confirm if the novel multiomic liquid chromatography–mass spectrometry (LCMS)-based technology would identify biomarkers in chickens, we used the analysis for the first time in chicken serum samples to identify essential biomarkers that were modulated due to the colonization of *C. jejuni* in broiler chickens. Furthermore, we investigated the effect of in-water supplementation of eugenol nanoemulsion on the host factors in the presence and absence of *C. jejuni*. 

## 2. Materials and Methods

### 2.1. Experimental Design

All the experiments were approved by the Institutional Animal Care and Use Committee of the University of Arkansas and recommended guidelines were followed for animal handling. For this study, we obtained 30 individual day-of-hatch broiler chicks (Cobb by-product breeder chicks) from a commercial hatchery. Chickens were weighed and randomly assigned to one of three groups [negative controls, positive controls (inoculated with *C. jejuni*), and treatments (supplementation of eugenol in *Campylobacter* colonized birds)]. Chicks were raised in floor pens with pine shavings with ad libitum access to feed and water throughout the trial period of 14 days based on National Research Counsil Nutrient’s requirement. The eugenol nanoemulsion was supplemented in the drinking water from day 0 for chicks in the treated group. The preparation of eugenol nanoemulsion was conducted as per the previous publication [11]. In addition, birds in both positive controls and treated groups were challenged with *C. jejuni* (250 µL/bird of 6 Log CFU/mL) on day 7. On day 14, blood serum was collected from five birds per group. Moreover, the confirmation of *C. jejuni* colonization was conducted by plating of cecal contents on *Campylobacter* line agar [14] as described previously [9].

### 2.2. Multi-Omics Analysis of Blood Serum

#### 2.2.1. Sample Handling and Preparation

Upon receipt, samples were inventoried and stored at −80 °C until further analysis. Samples were randomized and prepared using an automated liquid handling system (Opentrons Labworks OT-2). Multiple isotopically labeled recovery standards were added prior to the first preparation step for quality control purposes (QC). Samples were denatured with methanol, pH-buffered with ammonium bicarbonate, and chelated with EDTA. The sample protein was digested with 1:10 trypsin-to-protein by mass for 2 h at 37°C to generate LC-MS-amenable peptides. To remove the undigested matrix, the digest was precipitated with additional ethanol and acetonitrile, followed by centrifugation. The resultant supernatant extract was transferred and stored at 4 °C until further analysis.

#### 2.2.2. Liquid Chromatography Mass Spectrometry (LC–MS)

Samples were analyzed using a dual pump Thermo Vanquish liquid chromatography (LC) system coupled to a Thermo Scientific Q Exactive Plus mass spectrometer via electrospray ionization (ESI). The injected sample preparation was loaded onto a reverse phase column (Waters CSH-C18), and the flowthrough was diluted in-line prior to loading onto the HILIC column (Waters Z-HILIC). The columns were eluted sequentially using gradients composed of water, acetonitrile, and isopropanol modified with mobile phase additives, including formic acid and ammonium acetate. For quantification, the data was acquired using MS1 scans, and for identification, the data was acquired using data-dependent MS2 scans with a dynamic exclusion in both positive and negative ion modes. The mass analyzer was operated at 17,000 to 70,000 mass resolution, covering a scan range of 80 to 1200 m/z. The LC–MS Thermo RAW data was converted to open mzML format before data processing using ProteoWizard msConvert [15].

#### 2.2.3. Peptide Fractionation

For deep protein identification, peptides were fractionated using a high-pH reverse phase (Waters BEH-C18) and a water–acetonitrile gradient buffered with ammonium bicarbonate. The collected peptide fractions were dried down (SpeedVac) and resuspended in water acidified with formic acid.

### 2.3. Data Processing and Analysis

#### 2.3.1. Identification

Ions from the data-dependent MS2 data on pooled samples were identified by comparison to MS2 libraries composed of experimental and theoretical spectra. The Gallus gallus UniProt protein database was theoretically digested and fragmented for predicted MS2 ions (MSFragger FDR < 0.01). Theoretical lipid, MS2 spectra, were generated from fatty acyl chains and diagnostic head group ions (LipiDex, forward dot product > 0.5 and reverse dot product > 0.7). Metabolite and small molecule, MS2 spectra, were obtained from an experimentally acquired spectral repository (Compound Discoverer, match score > 0.8).

#### 2.3.2. Relative Quantification

The raw MS1 data were calibrated and quantified based on the retention times and m/z’s of identified ions. Relative quantification was extracted by summing peak intensities. Ions were filtered for minimum coverage across experimental samples (coverage > 50%) and absence in blank QC water injections. Outlier samples were rejected if correlation-based distance from other sample profiles was excessive. Sample profiles were statistically corrected for technical effects (run order). Multiple analyte forms of biochemicals were averaged to molecule-level relative quantification, e.g., multiple peptides and charge states were condensed into a single parent protein and multiple adduct forms of lipids were condensed into lipid species, etc.

### 2.4. Data Analysis

Statistical associations between biochemicals and the experimental variable (treatment group) were tested using linear regression. Multiple testing was addressed using false discovery rate (FDR) correction of nominal significance (*p* value). Differential associations were visualized using volcano plots and global data patterns were visualized with dimensionality reduction plots (hierarchical clustering, PCA). The functionality of the selected analytes was determined using UniProt.

## 3. Results and Discussion

### 3.1. Confirmation of C. jejuni Colonization in Broiler Chicks

We observed *C. jejuni* colonies in the selective agar plates after plating feces from the cloaca using cloacal swabs after two days of post-challenge as well as in the cecal contents at the end of the trial period (14 days). *C. jejuni* was not detected in the negative controls (chickens in the not-inoculated group) at the end of day 14. In the case of positive controls, an average *C. jejuni* colonization was ~6.5 Log CFU/g of cecal contents observed on day 14. The treatment with 0.125% EGNE reduced counts by ~2.5 Log CFU/g compared to positive controls. There were no significant differences in body weight gain among the negative control, positive controls, and treatment group.

### 3.2. The Effect of C. jejuni Colonization on the Profiles of Blood Serum in Broiler Chickens

We identified 1216 different analytes using Omni-MS multiomic technology in the blood serum of 14-day-old broiler chickens (Appendix A). The analytes were proteins (527), lipids (407), compounds (275), and inorganics (7). Several components significantly changed in the blood serum after *C. jejuni* colonization in the birds (Figure 1). For example, CREG1, creatinine, and 3-[2-(3-Hydroxyphenyl) ethyl]-5-methoxyphenol significantly upregulated after *C. jejuni* colonization. Protein CREG1 is responsible for neutrophil degranulation in chickens and acts as a biomarker for the inflammatory disorder [16]. BH3-interacting domain-death agonist (BID), the protein responsible for the activating apoptosis signaling pathway, also increased in *C. jejuni*-colonized birds. BID [17]. The increased levels of both CREG1 and BID suggested a disease condition at a molecular level. In addition, sphingosine, SP d18:1, HMGB3 (high mobility group protein B3), PC 11:0_26:1, phosphatidylcholines (PC) P-20:0_16:0, and PC 13:0_26:2 were downregulated in *C. jejuni*-colonized birds. Previously, decreased levels of sphingosine were found in association with the colonization of *Staphylococcus aureus* [18]. Sphingosine acts as a natural antimicrobial agent in the body. Similarly, HMGB3 is a multi-functional protein involved in the innate immune response by acting as a cytoplasmic promiscuous immunogenic DNA/RNA sensor. Plasmalogens play a crucial role in the defense against lipid oxidation [19]. The decreased expression of these biomolecules in chickens could indicate a reduced immunity level of birds colonized with *C. jejuni*.

The 5-hydroxyindole-3-acetic acid (HIAA) is a central metabolite of serotonin which was significantly upregulated in the blood serum of chickens supplemented with eugenol nanoemulsion (Figure 2). Previously, we reported that eugenol had an antibacterial action against *C. jejuni* in chickens and was found to modulate critical genes for attachment, colonization, and survival [10,11,12]. However, the functional mechanism of this essential oil against *C. jejuni* colonization in the blood serum of broiler chickens has not been elucidated yet. Earlier studies reported increased levels of several metabolites, such as 5-hydroxyindole-3-acetic acid, in the cecal contents of broiler chickens fed with probiotic dietary supplements [20]. The increase could be related to changes in the microbial metabolites in the gut because of eugenol supplementation. While comparing the serum metabolites in chickens free of *C. jejuni* and chickens supplemented with eugenol, we observed the modulation of several biological molecules (Figure 3). Eugenol supplementation significantly increased the level of 5-hydroxyindole-3-acetic acid, pyridoxal, taurine, indole-3-lactic acid (ILA), stachydrine, taurousodeoxycholic acid, monoolein, and methionine sulfoxide. These metabolites exhibited beneficial effects in the chickens via antioxidants and anti-inflammatory and antimicrobial functions. For example, taurine, taurosodeoxycholic acid, pyridoxal, and stachydrine enhance nerve growth in chickens [21]. Similarly, ILA, produced as a tryptophan metabolite, has anti-inflammatory properties. Ehrlich et al. [22] reported reduced inflammation of intestinal epithelial cells due to the protective action of ILA induced due to *Bifidobacterium longum* subsp. *infantis*. The observed downregulation of several plasmenylcholines is most likely due to the effect of *C. jejuni* colonization (Figure 1 and Figure 3) rather than the effect of eugenol, as these metabolites were not changed when compared between *C. jejuni*-colonized birds with and without eugenol treatment (Figure 2). Plasmalogens act as an endogenous antioxidant and are found to be elevated in human disease conditions [19]. The increased plasmalogens in the blood serum of broiler chickens challenged with *C. jejuni* could be due to the dysfunction of lipid oxidation in birds colonized with *C. jejuni*. Since we confirmed that the novel LC-MS-based technology is valuable in identifying potential biomarkers in chickens, it is crucial to perform further research with more time point analysis to provide in-depth functional mechanisms. 

## 4. Conclusions

*Campylobacter* colonization is not associated with any clinical signs in poultry, which creates a challenge for identifying positive flocks and implementing intervention strategies before carcass contamination occurs at slaughter. In the current study, using the novel LCMS-based multiomics technology, we identified the key biomarkers in the blood serum of broiler chickens associated with *C. jejuni* colonization and the effects of in-water supplementation of eugenol nanoemulsion on *C. jejuni*. We have identified some essential proteins and lipids involved in inflammation and immunity in the *C. jejuni*-colonized chickens and identified the beneficial effects of eugenol nanoemulsion in reducing *C. jejuni*. The identified biomolecules could serve as a biomarker for the differentiation of *C. jejuni* colonization in chickens. Further studies are essential to analyze the blood serum samples from multiple time points to identify other biomolecules involved during the colonization of *C. jejuni*.

## Figures and Tables

**Figure 1 animals-13-00559-f001:**
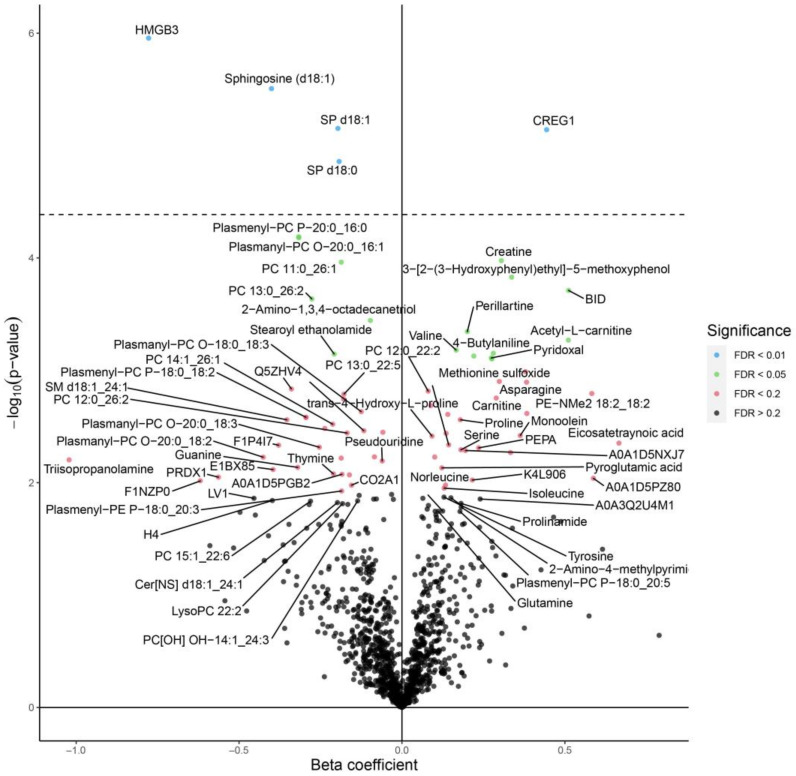
Volcano plot showing the differential expression of metabolites in the serum of broiler chickens challenged with *C. jejuni* on day 7 (positive control) compared to the unchallenged birds (negative controls). The serum samples were collected on day 14 and evaluated using Omni-MS multiomic technology. The level of significance was determined at FDR of 0.01 (blue dots) and 0.05 (green dots) in the volcano plot and was separated by the horizontal dotted line.

**Figure 2 animals-13-00559-f002:**
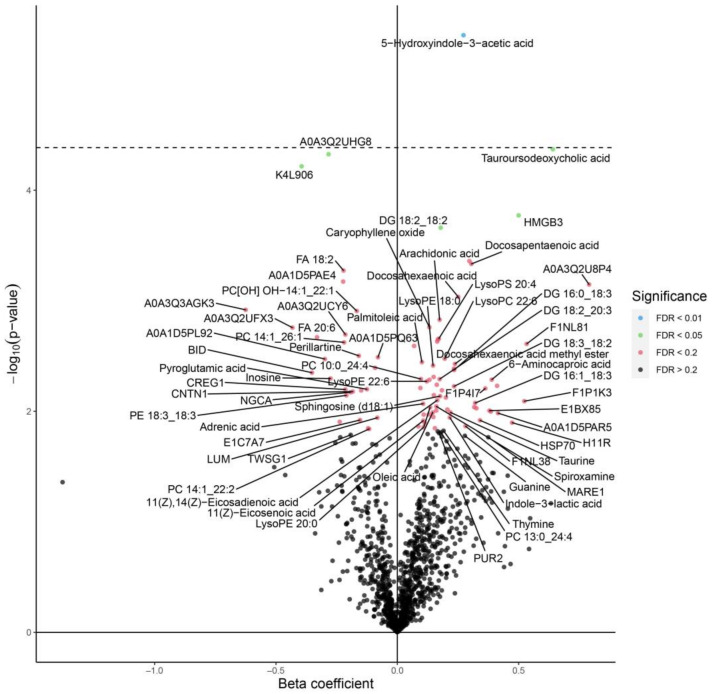
Volcano plot showing the differential expression of metabolites in the serum of broiler chickens supplemented with eugenol nanoemulsion (EGNE, *C. jejuni* challenge on day 7) compared to the positive control (no treatment, *C. jejuni* challenge on day 7). The serum samples were collected on day 14 and evaluated using Omni-MS multiomic technology. The volcano plot determined levels of significance at FDR of 0.01 (blue dots) and 0.05 (green dots) and was separated by the horizontal dotted line.

**Figure 3 animals-13-00559-f003:**
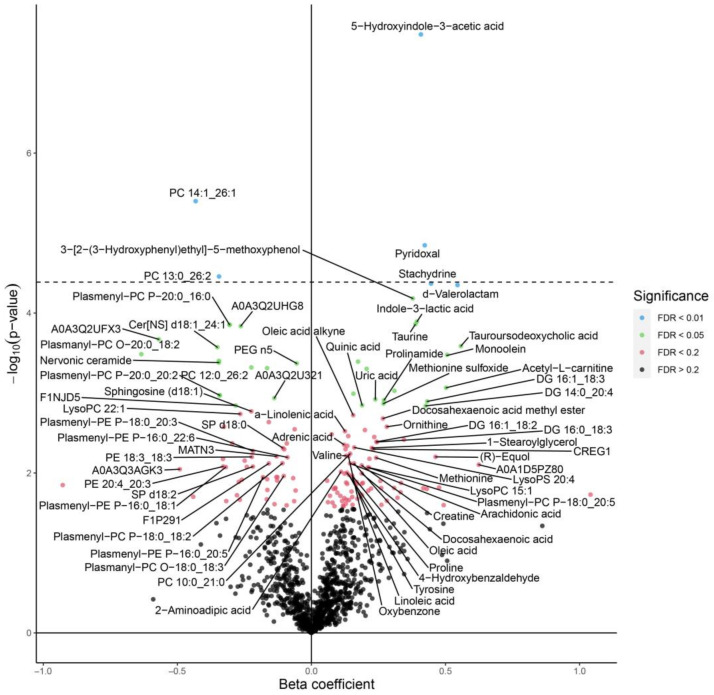
Volcano plot showing the differential expression of metabolites in the serum of broiler chickens supplemented with eugenol nanoemulsion (EGNE, *C. jejuni* challenge on day 7) compared to the negative control (no treatment, no challenge). The serum samples were collected on day 14 and evaluated using Omni-MS multiomic technology. The volcano plot determined levels of significance at FDR of 0.01 (blue dots) and 0.05 (green dots) and was separated by the horizontal dotted line.

## Data Availability

Raw data supporting the conclusions of this manuscript will be made available upon request.

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
