# Peer review of "A Multiomic Analysis of Chicken Serum Revealed the Modulation of Host Factors Due to Campylobacter jejuni Colonization and In-Water Supplementation of Eugenol Nanoemulsion"

_animals, 2023, doi:10.3390/ani13040559_

Round 1
Reviewer 1 Report
L41-42 please list the potential targets
L96 change "sample preparation" to "further analysis".
L118, check the spelling of "Proteowizard msconvert" and give the citation.
L150 change "p-value" to "P value"
L166 delete "in ceca"
L168 change "is" to "was"
L171 change "suggest" to "suggested"
check the tense in the results and discussion section
L217 change "are" to "was"
L239 “C.” should be italic.
check the referece 3, 5, 11
Author Response
Reviewer #1
We appreciate your comments and suggestions, and we have addressed them.
L41-42 please list the potential targets
Response:
Thank you for the suggestions, but due to the limited characters in the abstract, we cannot include the target list, but they are listed in the results and discussion section.
L96 change “sample preparation” to “further analysis”.
Response:
Changes have been made in the text.
L118, check the spelling of “Proteowizard msconvert” and give the citation.
Response:
The spelling was correct, however, we capitalized ‘W’ in Proteowizard and ‘C’ in msconvert.
The reference has been provided. Here is the proper spelling of the tool: “ProteoWizard msConvert”.
The citation below is included in the text:
Chambers, M. C., Maclean, B., Burke, R., Amodei, D., Ruderman, D. L., Neumann, S., ... & Mallick, P. (2012). A cross-platform toolkit for mass spectrometry and proteomics. Nature biotechnology, 30(10), 918-920.
L150 change “p-value” to “P value”
Response:
It is correct in the text.
L166 delete “in ceca”
Response:
It is deleted.
L168 change “is” to “was”
Response:
It is changed.
L171 change “suggest” to “suggested”
Response:
It is changed.
check the tense in the results and discussion section
Response:
The tense is corrected in the result and discussion section.
L217 change “are” to “was”
Response:
It is changed.
L239 “C.” should be italic.
Response:
It is italicized.
check the referece 3, 5, 11
Response:
All the references are corrected.
Reviewer 2 Report
Wagle et al. conducted an interesting study based on a multiomic analysis of chicken serum. However, in order to recommend acceptance of this paper, more work have to be done to improve clarity and biological interpretation.
Major concern:
Abstract: the need and aim of the study are concisely state. However, the link between the need and aim lacks clarity. All information is relevant.
Methods:
1. What guidelines were following for bird handling?
2. How were the animals sacrificed to prevent risk of contamination by the pathogen to the environment?
3. Food and water access details are missing?
4. Omics are easily affected by diet, age, time of day, gender. When were the blood samples collected, time of the day? Were all samples collected at the same time? What was the nutritional status of chicken during the collection time?
Results and discussion:
1. The findings are presented in a reasonable manner but connection to the aim and significance lacks clarity.
2. No work has been done to describe the impact of the infection on the weight of the birds, no inflammatory markers have been measured which are of paramount importance in infections.
3. Data for the confirmation of C. jejuni were missing.
4. All figures were poorly presented.
5. There are several linguistic flaws throughout the manuscript that the authors should address.
Minor concern:
There are many spelling and formatting problems, e.g.
1. Line 29: change "that enable" into "for".
2. Line 31: a space is needed before and after "=".
3. Line 50:the species name should be full at the beginning of a sentence.
Author Response
Reviewer #2
Wagle et al. conducted an interesting study based on a multiomic analysis of chicken serum. However, in order to recommend acceptance of this paper, more work have to be done to improve clarity and biological interpretation.
Response:
Thanks for the valuable comments and suggestions. All the concerns raised have been addressed in detail as follows.
Major concern:
Abstract: the need and aim of the study are concisely state. However, the link between the need and aim lacks clarity. All information is relevant.
Response:
The need and the aim have been extensively updated to provide clarity and connection.
Methods:
- What guidelines were following for bird handling?
Response:
The guidelines are developed by Institutional Animal Care and Use Committee of University of Arkansas. All the individuals involved with the animals used in this project were required to complete the CITI training module (“Working with the IACUC” for Researchers, Animal Technicians, Research Administration/staff and students working with animals) and are instructed in the humane care, handling, and use of animals, prior to any participation in the project. All the individuals responsible for animal studies have to comply with the Institutional Animal Care and Use Committee of University of Arkansas guidelines for proper handling, feeding, and management of birds to cause less pain and physiological distress during raising, euthanasia and disposal of remains.
- How were the animals sacrificed to prevent risk of contamination by the pathogen to the environment?
Response:
The animals were sacrificed in a post-mortem room. During sample collection, the abdominal cavity of birds were excised at minimum and the selected samples were dipped in the liquid nitrogen (for ceca) or kept in the enclosed box with ice to prevent environmental contamination. In addition, the carcasses were incinerated following sample collection. Appropriate biosafety protocols were followed by the personnel involved in the research and are adequately trained before being allowed to work with the foodborne pathogens.
- Food and water access details are missing?
Response:
The information had been provided under section 2.1 and we have highlighted it for your reference.
- Omics are easily affected by diet, age, time of day, gender. When were the blood samples collected, time of the day? Were all samples collected at the same time? What was the nutritional status of chicken during the collection time?
Response:
We understand Omics may be affected with various factors. All the samples were collected in the morning before feeding.
Results and discussion:
- The findings are presented in a reasonable manner but connection to the aim and significance lacks clarity.
Response:
Thank you for the comment and appreciate the opportunity to clarify it. We would like to reiterate that there are two objectives of this research 1) to explore the critical compounds that are modulated with the colonization of C. jejuni in chickens 2) to evaluate the effect of eugenol nanoemulsion on the host factors in the chickens colonized with C. jejuni. In order to match the objective with the results, at first, we have presented the data from the negative controls and then from positive controls. In addition, we compared the results among different groups and interpreted reasons behind the differences in expression of those biomolecules especially relating to chicken’s defense mechanism.
- No work has been done to describe the impact of the infection on the weight of the birds, no inflammatory markers have been measured which are of paramount importance in infections.
Response:
C. jejuni is a commensal in the chicken gut and it is well studied that colonization with C. jejuni doesn’t affect the body weight of chickens. In addition, we have recorded the body weight gain of each bird to see if there is any significant differences in body weight among the groups. We didn’t observe any significance difference. This statement has been included in the revised manuscript (Section 3.1). Regarding the inflammatory markers, we have utilized a highly sensitive multi-omics technology to evaluate the expression of all types of proteins, lipids, compounds and inorganics in this research. Overall, we detected 1216 different analytes in the chicken serum and many of these are related to inflammatory markers. Due to variation in functions of these, our results focused on the analytes that were significantly changed among the groups. In addition, we believed that the analytes that were not changed among the groups doesn’t confer any useful information.
- Data for the confirmation of C. jejuni were missing.
Response:
All the birds in positive controls and treatment group were positive to C. jejuni based on plating of cecal samples. In addition, the birds in negative controls were negative to C. jejuni. The colonization data along with other information are part of another ongoing publications and in order to remove duplication of information between the publications, we decided not to share that data in this research note. In addition, we are bound by the length of this research note based on the guidelines of MDPI.
- All figures were poorly presented.
Response:
We updated the figures to enhance the quality.
- There are several linguistic flaws throughout the manuscript that the authors should address.
Response:
We have made extensive changes to avoid linguistic flaws.
Minor concern:
There are many spelling and formatting problems, e.g.
- Line 29: change “that enable” into “for”.
Response:
That sentence is rephrased.
- Line 31: a space is needed before and after “=”.
Response:
Space is included.
- Line 50:the species name should be full at the beginning of a sentence.
Response:
The full name ‘Campylobacter’ is included.
Round 2
Reviewer 2 Report
Most problems have been solved, but data for the confirmation of C. jejuni were still missing, which is very important. I recommended the missing data should be included in the MS or supplimentary files.
Author Response
Most problems have been solved, but data for the confirmation of C. jejuni were still missing, which is very important. I recommended the missing data should be included in the MS or supplimentary files.
Response:
Thanks for accepting all the other information. We have included the data for C. jejuni confirmation under section 3.1 and highlighted it in blue.
“C. jejuni was not detected in the negative controls (chickens in the not-inoculated group) at the end of day 14. In the case of positive controls, an average C. jejuni colonization was ~ 6.5 Log CFU/g of cecal contents observed on day 14. The treatment with 0.125% EGNE reduced counts by ~ 2.5 Log CFU/g compared to positive controls.”